# Faithful Stewards of God's Creation? Swedish Evangelical Denominations and Climate Change

**Karin Edvardsson Björnberg** [1,*] and **Mikael Karlsson** [2]

1 Division of Philosophy, KTH Royal Institute of Technology, SE-100 44 Stockholm, Sweden
2 Department of Earth Sciences, Uppsala University, SE-752 36 Uppsala, Sweden; mikael.karlsson@geo.uu.se
* Correspondence: karine@kth.se

**Abstract:** Studies from the United States (U.S.) show that opposition to climate policy is strong among some Christian groups, especially White evangelical Protestants. Much of this opposition is channelled through organisations such as the Cornwall Alliance, which argue against climate measures on religious, economic and what they claim to be science-based grounds. In the present study, we investigated to what extent these convictions were present among Swedish evangelical denominations. Representatives from the Evangelical Free Church, the Pentecostal Alliance, the Swedish Alliance Mission, and the Seventh-day Adventist Church were interviewed to identify the denominations' views on the scientific underpinnings of climate change and the moral implications of climate policy. Our data show that the denominations' views differ markedly from those expressed by climate-oppositional evangelical groups in the U.S. The denominations held homogenous views on the legitimacy of climate science, expressed a clear biblical mandate for climate policy based on the notion of human stewardship, and believed that climate change was inextricably linked to poverty and, thus, had to be addressed. Our results point to the need for further studies on the factors behind acceptance and denial of climate science within and between faith-based and other communities in different countries.

**Keywords:** evangelical; climate change; environment; creation care; climate denial; Sweden; United States





## 1. Introduction

In the United States (U.S.), White evangelical Protestants are among those who are most sceptical of climate science and least willing to support climate policy (Shao and McCarthy 2020; Veldman et al. 2021). Survey work has consistently shown that White evangelicals dismiss the anthropogenic origin of climate change to a larger degree than other faith groups (Jones et al. 2014; Leiserowitz et al. 2015). In one study, when asked whether they believed in anthropogenic global warming, only 28% of those who identified as White evangelicals gave a positive answer, in comparison to White mainline Protestants (41%), White Catholics (45%), Black Protestants (56%), and Hispanic Catholics (77%) (Pew Research Center 2015). White evangelicals are also less concerned about the impacts of climate change on humans, societies and the environment (Jones et al. 2014). Well-funded organizations affiliated with the Christian Right continue to spread doubt about climate science within the U.S. evangelical community, prompting key leaders to withdraw their support for climate policy (Perez Sheldon and Oreskes 2017). The strong links that exist between the climate-oppositional evangelical groups and high-ranking U.S. Republican Party members mean that the former have a real opportunity to influence the country's political agenda (Bean and Teles 2015; Veldman et al. 2021).

However, the evangelical community is in no way lost to the climate-oppositional movement. Many evangelical leaders and scholars, including Richard Cizik of the U.S. National Association of Evangelicals (NAE), have publicly defended climate science and argued for climate policy, drawing political and theological strength from the evangelical

'creation care' movement of the 1990s (Wilkinson 2012). Evangelical organisations, such as the Evangelical Environmental Network (EEN) and the Evangelical Climate Initiative (ECI), have mobilised evangelicals not only around climate change but also other environmental issues, such as pollution and biodiversity loss. Studies on lay evangelicals' beliefs and concerns for the environment have also confirmed that a diversity of positions exist within the U.S. evangelical community, especially within the younger generation, suggesting that the relationship between evangelicalism and climate denial is far from unambiguous (Smith and Leiserowitz 2013; Smith et al. 2018; Lamb et al. 2019).

Most research on evangelicalism and climate science and policy has focused on U.S. denominations. This is unsurprising considering the large percentage of evangelical Christians in the U.S. and their significant influence on national politics (McCammack 2007; Steensland and Wright 2014; Bean and Teles 2015). However, evangelicalism is a visible political force in many other countries too, not least in developing countries. Studies of the relationship between religious belief and environmental opinion have suggested there may be significant cross-country differences in how environmental and climate policy issues are framed and assessed (Hagevi 2014; Kempf 2020). Hagevi (2008) showed that individuals who attended religious meetings in Sweden tended to support environmental policy measures to a larger degree than their secular counterparts. These and similar findings point towards the need to investigate the relationship between evangelicalism and climate science and policy in different countries. Such studies could help to contextualise the phenomenon of climate denial and, by extension, shed light on the factors behind faith-based climate opposition in countries such as the U.S.

In this article, we analyse the views of four Swedish evangelical denominations on the scientific underpinnings of climate change and the moral implications of climate policy: Evangelical Free Church (*Evangeliska frikyrkan*), Pentecostal Alliance of Independent Churches (*Trossamfundet Pingst*), Swedish Alliance Mission (*Svenska Alliansmissionen*), and Seventh-day Adventist Church (*Sjundedags Adventistsamfundet*). Through desk-based research of published policy documents and interviews with representatives of the four denominations, we explore how the climate change issue is framed within a Swedish evangelical context: What are the denominations' views on the arguments voiced by climate-oppositional evangelical groups, foremost in the U.S.? How do the denominations frame the climate issue themselves, and how do they justify their efforts (or lack of efforts) from a biblical point of view? By addressing those questions, we hope to contribute to the sparse corpus of qualitative studies investigating the relationship between evangelical faith and attitudes to climate science and policy outside of a U.S. context.

In Section 2, we describe how the climate issue has been framed by the evangelical community in the U.S., focusing on the arguments put forward by evangelical climate-oppositional groups such as the Cornwall Alliance. This summary description will give the main lines of argument against which the findings of our investigation can be analysed.[1] In Section 3, we describe the data collection and analysis methods. In Section 4, we report the main results from the interviews. In Section 5, we discuss the results drawing on literature and previous studies in the field, and we identify issues that need further investigation.

In this article, we adhere to Bebbington's (1989) definition of evangelicalism (the 'Bebbington quadrilateral'), which identifies four distinguishing features: *Biblicism*, that is "a high regard for and obedience to the Bible as the ultimate authority"; *conversionism*, defined as "the belief that lives need to be transformed through a 'born-again' experience and a life long process of following Jesus"; *crucicentrism*, defined as "a stress on the sacrifice of Jesus Christ on the cross as making possible the redemption of humanity"; and *activism*, defined as "the expression and demonstration of the gospel in missionary and social reform efforts" (NAE 2022).

## 2. Climate Science and the Evangelical Community

A significant body of literature from the U.S. shows that environmental and climate concern is particularly low among White evangelical Protestants. There is an ongoing debate about which factors can help explain this finding, specifically whether the resistance to environmental and climate policy among evangelicals can most accurately be explained by reference to political orientation (including values and attitudes towards collective action), religious beliefs, or a combination of both (Veldman et al. 2021). A significant number of studies have confirmed that political conservatives tend to be less concerned about the environment than their liberal counterparts (Dunlap et al. 2001; McCright and Dunlap 2010, 2011; Antonio and Brulle 2011; McCright et al. 2014). Since many evangelicals, at least in the U.S., are politically conservative, it therefore comes as no surprise that resistance to environmental and climate policy is particularly prevalent within this group (Newman et al. 2016).

Veldman (2019) argues that evangelical climate denial in the U.S. cannot be understood in isolation from the socio-historical context in which it developed. For an extensive period of time after the nation was founded, evangelicals had considerable political influence through their key positions in society. However, around the turn of the twentieth century the evangelical community split into different factions, which diminished its socio-political influence. It was not until the last decades of the twentieth century that conservative evangelicals, increasingly concerned about the spread of secular humanism and liberal theology, were able to align themselves with non-religious conservatives, 'minimal state' proponents, industry representatives, and, at a later stage, climate change sceptics, to form a coalition powerful enough to influence the political agenda. By the end of the twentieth century, Veldman (2019) argues, the war against secular culture fought by evangelicals alongside these secular actors, had contributed to shaping a specific evangelical identity that not only encompassed the social and theological values traditionally held by the evangelicals themselves, but also the political and economic values promoted by their secular allies, such as increased consumer choice and free-market policy solutions (see also Kearns 2007). Hostility towards (environmental) regulations was part and parcel of what the secular allies brought to the table; hence, it was incorporated into evangelical identity.

Although most studies conclude that political orientation is the strongest predictor of climate opposition (see further in Edvardsson Björnberg et al. 2017), there is evidence to suggest that religious belief may also shape the environmental attitudes and actions of evangelicals (see Veldman et al. 2021 for a discussion). As noted by Veldman et al. (2021), there is currently a noticeable lack of qualitative studies investigating some of the possible religious drivers of climate opposition. In the literature, biblical literalism is sometimes pointed out as a factor that contributes to increased levels of climate denial (Guth et al. 1995; Smith and Leiserowitz 2013; Arbuckle and Konisky 2015; Morrison et al. 2015; Schwadel and Johnson 2017). Biblical literalism typically encompasses conservative eschatology and dispensational theology, confirming the idea that humans have dominion over nature, and ideas about God's sovereignty and continuous intervention in the world, all of which, arguably, could justify assigning a lower priority to environmental issues (Guth et al. 1995; Barker and Bearce 2013; Kilburn 2014; Peifer et al. 2016). However, further studies into the theological drivers of climate opposition are required to better understand how religious climate-oppositional groups position themselves and are able to sell their messages. In the remaining paragraphs of this section, the theological bases for climate activism and opposition are briefly explored. First, the eco-theological origins and arguments of the evangelical 'creation care' movement are briefly outlined. This summary rests heavily on Wilkinson (2012), to which the reader is referred for a more in-depth report. Then, the climate-oppositional arguments used by the evangelical environmental and climate counter-movement are summarized. These arguments are discussed in more detail in Edvardsson Björnberg et al. (2020). Together, these two sections provide a background description against which the findings of our investigation can subsequently be analysed.

*2.1. The Evangelical 'Creation Care' Movement*

The eco-theological origins of the creation care movement date back to the late 1960s and early 1970s when Francis Schaeffer, Calvin DeWitt, Loren Wilkinson and others began to formulate a distinctly evangelical response to White's (1967) seminal argument that the traditional Western Judeo–Christian worldview was at the root of the current environmental crisis. Among the core ideas they defended were that the environment, being God's creation, has final value, as opposed to value only in relation to the satisfaction of human well-being; that human dominion expressed in Genesis 1:27–28 should be understood as a call to responsible stewardship rather than permission for humans to use the environment for whatever purposes they see fit; that evangelicals should extend their concerns beyond individual salvation and toward societal engagement; and that environmental destruction has disproportionately negative impacts on poor people, and is therefore of great moral concern for evangelicals (Simmons 2009; Wilkinson 2012).

In the 1980s and 1990s, the ideas and principles endorsed by the early evangelical eco-theologians were further developed and eventually translated into political action. In 1993, the U.S. Evangelical Environmental Network (EEN) was established "with the aim of offering a uniquely evangelical alternative to other Christian environmental groups" (Wilkinson 2012, p. 19). A year after its foundation, the EEN adopted "An Evangelical Declaration on the Care of Creation", which was signed by around 150 representatives of the U.S. evangelical establishment (EEN 1994).

As political awareness of climate change increased at the turn of the millennium, the EEN and associated actors further developed their pro-environmental positions through a number of statements on climate change. In 2006, the Evangelical Climate Initiative (ECI), a network of evangelical church leaders and organisations, issued "Climate Change: An Evangelical Call to Action" (ECI 2006). With the establishment of ECI, Wilkinson (2012, p. 25) concluded that, "climate care was enshrined in the evangelical establishment", although it was far from being a consensus view.

Perez Sheldon and Oreskes (2017) identify two biblical bases for Christian (evangelical) environmentalism: natural theology and the moral call for social justice. Natural theology advocates the belief that God reveals himself in the natural world and that humans, therefore, have an obligation to study, revere and, importantly, take care of it. Hence, the preservation of wild spaces and biodiversity is encouraged, since it constitutes one way of showing "deep wonder and love of God's handiwork in Creation" (Perez Sheldon and Oreskes 2017, p. 360). This contrasts with the moral call for social justice, according to which evangelical Christians have an obligation to protect the environment, not so much as an act of love of God's handiwork, but as a means of assisting the global poor. Although both theological bases were elaborated on by the early evangelical eco-theologians and are referenced in the 1994 EEN declaration, it is clear from the ECI 2006 Call to Action that, for evangelicals, social justice remains the most central justifications for engaging in climate advocacy (Perez Sheldon and Oreskes 2017).

The EEN's and the ECI's pronounced efforts to engage the U.S. evangelical community in the environmental debate have led some scholars to suggest that we are currently witnessing a "greening of evangelical Christianity" (Hitzhusen 2007; Wilkinson 2010, 2012). This is further supported by the recent efforts of Young Evangelicals for Climate Action (YECA), an official ministry of EEN aiming to mobilize young evangelicals to step up on climate action (Lamb et al. 2019). Among the actions taken by the YECA are the so-called "Climate Testimonies", recorded personal stories of how individual members came to care about climate change, that are posted with the aim to engage the evangelical community. This is also supported by the efforts of the National Association of Evangelicals (NAE) to include creation care and climate change considerations into the association's policy work. In the 2018 call to civic responsibility, the NAE identified climate change as "a threat multiplier" that increases damage to creation and has grave implications for the poorest, a clear recognition of the urgency of the climate issue.

However, the argument that evangelical Christianity is presently witnessing a break point in terms of climate change commitment is contested in the literature. Although key evangelical institutions have taken up environmentalist ideas in recent years, some scholars conclude that such ideas and principles are yet to be adopted by the broader evangelical community (Konisky 2018; see also Clements et al. 2014; Carlisle and Clark 2018).

*2.2. The Evangelical Climate Counter-Movement*

In the mid-2000s, as evangelical climate care ideas were beginning to gain ground, a renewed wave of opposition was launched by a group of conservative evangelicals who had formed close bonds with free-market proponents, secular climate change sceptics and key members of the U.S. Republican Party (Bean and Teles 2015; Ronan 2017). The Cornwall Alliance, a coalition of predominantly evangelical leaders and scholars, established support for 'biblical earth stewardship' in the early 2000s, one of the driving forces behind this counter-movement. Its 2006 open letter to the ECI signatories questioned the economic and moral justifications of emissions reduction, using theological, political, economic, and what the group claimed were scientific, arguments (Cornwall Alliance 2006). The scripture-based arguments against climate action put forward by the Cornwall Alliance and associated actors (henceforth, the 'Cornwallists') effectively capture the ideological core of the evangelical environmental and climate counter-movement. Therefore, together with the eco-theology of the creation care movement, they can be used as a discursive frame for inquiries into the theological drivers of evangelical climate activism and opposition outside of a U.S. context. The ideological core of the Cornwallists can be captured through a number of scripture-based arguments against climate action summarized below (Edvardsson Björnberg et al. 2020).

*Anti-paganism argument.* A key argument put forward by the Cornwallists is that it is wrong to put the environment and the climate system at the centre of one's attention—for example, by working intensely towards greenhouse gas emissions reduction—since it amounts to serving "the creature rather than the Creator" (Danielsen 2013; Daley Zaleha and Szasz 2015). Put differently, it is an impermissible form of paganism, or pantheism, that denies the absolute sovereignty of God. Although the Cornwallists recognize that humans have a responsibility to care for God's creation, the actions proposed by the EEN and the ECI, among others, manifestly cross the line between what is prudent and called for and what is explicitly forbidden by the Bible.[2]

*Enrichment argument.* Related to the ontological separation of the Creator and the creation is the idea of a hierarchical ordering of the creation and the elevated position of humans within it. As bearers of God's image, humans have a key role to play within the creation. This key role involves, among other things, a right and a duty to exercise dominion over the rest of creation. In many Christian traditions, this is interpreted as a duty to exercise stewardship, to care for the environment and other living creatures. However, the Cornwallists interpret the dominion injunction not so much in terms of conservation or preservation, but rather as a call from God to actively transform the natural world as to enrich creation (Beisner 1997; see McCammack 2007; and Wardekker et al. 2009 for a discussion). Only by enriching creation can humans re-establish the long-lost Garden of Eden, in other words, turning wilderness into a garden. According to this view, restricting welfare-generating activities through, for instance, the enactment of emissions reduction is wrong since it conflicts with a divine task assigned to humans by God.

*Omnipotence argument.* Another argument put forward by the Cornwallists is that worrying about climate change makes little sense, since God is in control (Nagle 2008; Hempel et al. 2014). The "Earth and all its physical and biological systems are the effects of God's omniscient design, omnipotent creation and faithful sustaining" (Cornwall Alliance 2013; see also Beisner 2005) and is, therefore, capable of withstanding any physical interferences. Suggesting otherwise could actually be considered an affront to God.

*Lack of moral relevance argument.* A fourth argument against climate action endorsed by the Cornwallists is that climate change is morally insignificant in comparison to issues such

as salvation and saving souls, fighting abortion, gay marriages, human trafficking, and not least, global poverty (Danielsen 2013). Frequently, emissions reduction is portrayed as compromising international efforts to combat poverty, making it more difficult for poor people in the Global South to develop and bring themselves above the poverty line.

*Cost–benefit argument.* Related to the previous argument is the idea that the costs of climate action grossly exceed the expected benefits, and for this reason alone ought to be dismissed (Beisner 2005; Cornwall Alliance 2000). Thus, the Cornwall Alliance (2008) argues that climate change is "driven primarily by variations in non-human forces", the contribution of humans to climate change is highly uncertain, and the benefits of continued fossil fuel use outweigh the risks involved.

*'End times' argument*: A final argument attributed to conservative evangelicals states that there is little point in mitigating climate change or worrying about it since, according to the Bible, the Earth will be destroyed at some point in the near future anyway (Nagle 2008; Simmons 2009; Barker and Bearce 2013; Daley Zaleha and Szasz 2015). To the authors' knowledge, eschatological beliefs are not explicitly referred to in any policy documents issued by the Cornwall Alliance, nor does there seem to be solid empirical support for the idea that end time beliefs have a real impact on evangelicals' willingness to support climate action (Veldman et al. 2021; see also Hitzhusen 2007 and Veldman 2019). However, since there are some discussions in the literature of whether such beliefs could actually help to explain the hostility towards climate policies within the evangelical community (Guth et al. 1995; Barker and Bearce 2013; Haltinner and Sarathchandra 2021), we decided to include them in our analysis.

## 3. Materials and Methods

### 3.1. Denominations Selection

Studies on the climate change beliefs and attitudes of evangelicals presuppose that it is possible to identify the individuals and institutions that are said to be 'evangelical'. As noted by Veldman et al. (2021), evangelicals can be identified by reference to certain beliefs (e.g., the 'Bebbington quadrilateral'), adherence to the evangelical tradition though membership of certain churches or denominations, or though self-identification. In this study, all of these strategies were used to some extent in our selection of denominations. Denominations were initially identified using either formal affiliations with the Swedish Evangelical Alliance or satisfaction of the 'Bebbington quadrilateral' as criterion. Denominations known to have fewer than 2000 registered members were excluded, since we assumed it was less probable that they would have well-developed policies on climate change. Four denominations with a sufficiently large membership were identified as being affiliated with the Swedish Evangelical Alliance: Evangelical Free Church, Pentecostal Alliance of Independent Churches, Salvation Army and Word of Faith Movement/Word of Life.[3] Three more denominations were identified using Bebbington's definition of evangelicalism: Swedish Alliance Mission, Equmenia and Seventh-day Adventist Church. In the case of the Seventh-day Adventist Church, which could arguably be defined as Adventist rather than evangelical, the denomination was also asked whether it would identify as evangelical, something that the denomination's representative agreed to.

These seven denominations were emailed with an open invitation to participate in the study. The email contained information about the study's purpose and methods, the potential risks and benefits of participation, and the research subjects' rights according to the Swedish Ethical Review Act (2003) and the General Data Protection Regulation (EU) 2016/679. Four denominations responded positively, and they form the paper's empirical basis (Table 1).

**Table 1.** Statistics about the selected denominations in the present study.

| Name of Denomination | Number of People Served [a] | Website |
| --- | --- | --- |
| Evangelical Free Church | 44,674 | https://efk.se, accessed on 22 February 2022 |
| Pentecostal Alliance | 114,953 | https://www.pingst.se, accessed on 22 February 2022 |
| Swedish Alliance Mission | 18,028 | https://alliansmissionen.se, accessed on 22 February 2022 |
| Seventh-day Adventist Church | 3289 | https://www.adventist.se, accessed on 22 February 2022 |

[a] = Based on statistics from the Swedish Agency for Support to Faith Communities, available online (in Swedish): https://www.myndighetensst.se/bidrag/organisationsbidrag/bidragsgrundande-statistik.html (accessed on 22 February 2022).

### 3.2. Desk Research

The denominations' websites were carefully scrutinised to gain an overview of their views on climate change. Policy documents on climate change and the environment, such as declarations of intent and open letters, were identified by using the term 'climate change' (*klimatförändring/-en/-ar* in Swedish) in each website's search area. All relevant documents were downloaded and read. The participating denominations were also asked to provide the researchers with additional relevant material, which they did.

### 3.3. Semi-Structured Interviews

The desk-based study of the policy documents quickly revealed that additional data would be needed to answer the overall research questions. To gain in-depth information about the participating denominations' views and standpoints, semi-structured interviews were conducted. In all, seven persons who held formal positions within the respective denominations, typically responsible for the denominations' environmental or sustainability work, participated in the interviews. The interviews followed a template and were conducted after an application for ethical vetting was approved by the Swedish Ethical Review Authority.[4] The interview template was based on the information gathered from the desk-based review of the policy documents and structured along the identified arguments put forward by evangelical climate-oppositional groups outlined above (see Appendix A). Hence, it included questions targeting the *anti-paganism*, *enrichment*, *omnipotence*, *lack of moral relevance*, *cost–benefit*, and '*end times*' arguments as well as two general questions about the denominations' views on the conclusions drawn in the IPCC (Intergovernmental Panel on Climate Change) reports and the Biblical stewardship injunction. The template was communicated to the denominations beforehand to give them the opportunity to prepare for the interviews. The interviews lasted for 45–60 min and were recorded and transcribed. The transcribed interviews were read and approved by the interviewed denominations.[5] During this process, the interviewed representatives were allowed to make adjustments to the transcribed interviews. Only a few minor adjustments were made by the interviewees during this process. The analysis of the transcriptions was guided by the theological themes identified above, and common patterns in the respondents' comments were searched for.

## 4. Results

During the interviews, several informants emphasised the difficulties of giving generally applicable answers to questions on which local parishes may have divergent views. The evangelical movement is an anti-authoritarian and heterogeneous movement, and local parishes traditionally enjoy a great degree of autonomy and freedom in developing their interpretations of the evangelical credo. In contrast to, for instance, the Catholic Church, there are few policy documents and relatively little guidance on how theological

and practical matters should be addressed by local parishes. These caveats must be kept in mind when reading the sections below.

*4.1. General Climate Science Views*

The data showed that the interviewed denominations were homogeneous in their views on the legitimacy of climate science. In general, the denominations had adopted few, if any, policies specifically addressing climate change. However, the interviewed representatives acknowledged that climate change was an increasingly prioritised policy issue within their respective denominations. This was partly attributed to the fact that the denominations were involved in the development of co-operations in countries where the effects of climate change were already noticeable, and partly because young members of the denominations were actively pushing for climate change to be observed in the denominations' activities.

None of the informants had listened to any pronouncement in which the central tenets of climate science—climate is changing, human activities are the main cause, and the impacts of climate change will be predominantly negative—were denied. However, the Seventh-day Adventist Church's representative said that, although human activities can rightly be seen as the main cause of the changes in climate we are currently experiencing, other forces come into play as well. As a result of the Fall, the informant argued, humans have given Satan (evil) permission to reside in the world. Climate change is part of his deeds; therefore, humans cannot be said to be solely responsible for what is happening. At the same time, the informant stressed that this fact did not in any way relieve humans from the responsibilities they have to care for God's creation, including the climate system.

Several informants acknowledged that the assembly members' attitudes to climate change varied with age and that older members were generally more sceptical of climate science than younger members. The Swedish Alliance Mission's representative stated:

> "Once there was this elderly gentleman who said to me: 'Why should I believe in science when science does not believe in God?' That was sufficient reason for him to ignore everything to do with IPCC. So those people definitely exist."

That climate change received relatively little attention within the denominations until recently was explained, not so much by reference to climate science denial but by a widespread disinterest among the assembly members. One of the Pentecostal Alliance's informants stated:

> "I don't think there has been any science denial really; it has more to do with theology. From a theological point of view other issues, spiritual issues, have been prioritised."

Several denominations were well aware of the situation in the U.S. at this time, with the Trump administration and powerful representatives of the evangelical community denying basic climate science. When asked about what factors they believed could help explain the present situation, the informants gave somewhat different answers. One of the representatives from the Evangelical Free Church elaborated:

> "I don't know that much about the U.S. but unfortunately the issue has been connected with Lynn White, 1967, who has become very controversial and who divided people into different camps. [ ... ] There is also the issue of Democrats versus Republicans. In the end, for some Americans it's simply the case that if you're evangelical then you're also a Republican, that's how you're supposed to vote. And you will then be against abortion and you will be against gay marriage. And everybody who's interested in environmental issues is a Democrat. This division does not exist in other parts of the world."

The other representative agreed, adding that the climate issue had become an identity marker for many American evangelicals. A different set of identity markers was used within the Swedish evangelical movement, the informant argued, which could help explain why climate change was a much less stigmatised issue.

The Swedish Alliance Mission's representative gave a somewhat different explanation for why American evangelicals were unwilling to support climate mitigation, drawing on the history behind the nation's foundation:

> "I think the main problem is that if you read the Bible through a nationalistic lens. You see the U.S. as God's promised land [ ... ]. There are significant undertones in how the nation was founded and in how various declarations were formulated. If you read the Bible through that lens, then you may very well find arguments for overusing the planet's resources. It is your God-given right as an American or as a member of God's chosen people."

### 4.2. Stewardship vs. Paganism

When asked whether care for the environment (including climate change) could go too far, thus bordering on paganism, several informants answered that it could well be the case but only if the motives behind one's actions were of the wrong kind. As long as humans remember that the ultimate reason why they should care for the environment is that it is God's creation, and that by caring for it they fulfil his intentions, there is little to worry about. The Swedish Alliance Mission's representative stated:

> "Within the Swedish Alliance Mission, we have always focused on Jesus Christ, on evangelisation. If you were to talk about environmental issues in a way that does not include those issues, then people would react. But as long as we weave those things together and are clear about why we do this, it's because we focus on Jesus Christ, then we're on the right track."

However, the Seventh-day Adventist Church's representative cautioned that, even for devout Christians, it is possible to lose one's focus. As an example, the informant mentioned the hypothetical case of an Adventist becoming so preoccupied with his/her own bodily and mental health that s/he loses sight of everything else, perhaps even disregarding other people's feelings as a result of his/her ambitions. In such a situation, where keeping healthy becomes the most important life goal, it is not wrong to talk about paganism or idolatry. In a similar fashion, the informant argued, caring for climate change could go too far, for example if one disregards the rights of certain groups of people in order to protect the climate system.

### 4.3. The Dominion Injunction—To Preserve or Enrich?

When asked about what the idea of stewardship involves, the informants again gave fairly similar answers. The Seventh-day Adventist Church's representative stated:

> "As Christians and as members of the Adventist church, we believe that God has created the world. He has a plan. And we have a key role to play in this world, as parts of his creation but also as being situated somewhat above the creation, as stewards of the creation. To take responsibility, to govern and lead."

The Swedish Alliance Mission's representative elaborated on the difference between making use of and overusing the environment:

> "There is a section in Genesis 1 saying something like 'to keep and till'. And there is somewhat of a difference between using and overusing. We are indeed entitled to use whatever God has created. And there is also the idea that we are God's 'co-creators' [ ... ], and there is the idea about the Fall, that humans somehow fail to take their roles as stewards seriously, to exercise constraint. They fail to put an end to things, to say 'this is enough, I don't need any more'. And that's the cause of this overexploitation."

The Swedish Alliance Mission's representative pointed to two reasons why overexploitation of the environment is problematic: first, the Genesis creation narrative tells us that God created the Universe, and he saw that it was good. Importantly, it was good even before man was created. Thus, the informant concluded that the creation must ultimately be valuable. Second, the informant added, there is a human dimension to the issue. Jesus tells us to love our neighbours as we love ourselves. Therefore, if the overexploitation of nature turns out to have negative consequences for people's well-being, today or in the future, it must be stopped.

When asked whether they had encountered the Cornwallist idea that as God's stewards, humans have a responsibility to not merely preserve but enrich creation by transforming untouched nature into a garden (for example, by burning fossil fuels), all informants argued in the negative. Several informants connected the idea to what they called 'prosperity theology', which was not widespread within their denominations.

### 4.4. God's Omnipotence

When asked whether God's omnipotence could give humans a reason to refrain from taking mitigation measures, the informants again gave uniform answers. They agreed that God has the power to intervene in the world and put things right. Most informants interpreted this as something that would happen at the end times when Jesus returns, and God intervenes to save his creation. One of the Pentecostal Alliance's representatives argued:

> "You could give a positive answer to this question, but then it's like a 'total makeover', I mean God will redeem his creation, similar to when he created it. Bam! Like that. But if you think that he will all of a sudden put his hand and then the whole Antarctica will freeze. The tundra, or permafrost, will just . . . no, I don't think so."

However, all informants agreed that even if God is omnipotent in the sense that he has the power to redeem his creation at any given time, that does not give humans a reason to doubt climate science. In the informants' views, humans have a responsibility to care for God's creation as part of their stewardship role. If they do not take this responsibility seriously, then they do not live their lives according to God's will. One of the Pentecostal Alliance's representatives argued:

> "God has given us a responsibility for the times we are living in. It's not just about the environment but everything, all aspects of life. How we care about other people, peace [ . . . ]. In all aspects of life, he has given his flock, those who believe, a responsibility to care for the creation. [ . . . ] So just doing whatever you feel like doing, or thinking or acting according to your own impulses, that's not obeying God's will. You will then do something wrong."

One of the Evangelical Free Church's representatives emphasised:

> "Traditionally, we are an eschatological movement heavily influenced by the belief in Jesus Christ's second coming. [ . . . ] This has affected people's willingness to engage in environmental issues, because there is a belief and a tradition saying that no matter what happens, God is in control. And for sure, what we're witnessing is the world developing in an evil direction, but God will fix things anyway. It doesn't give us reason not to engage in environmental issues, but I still believe it has had a dampening effect."

### 4.5. Moral Relevance of Climate Change

When asked how relevant the climate change issue is from a moral point of view in comparison to other, perhaps more traditional, moral issues, such as abortion, gay marriage and poverty alleviation, the informants again gave rather similar answers.

As indicated above, several informants acknowledged that climate change was not an issue that had traditionally gained attention within the denominations' work, although this was about to change as young people pushed for the issue and the impacts of climate change became gradually more acute in the denominations' international development work. Various explanations were given for why climate and environmental issues had previously not been considered key to the denominations' work. A possible explanation brought up by one of the representatives of the Pentecostal Alliance related to the traditional Western body–soul dualism inherent in the denomination's theology:

> "Our theology has been influenced by the ancient Greek body–soul dichotomy, where material, earthly, aspects have been considered less valuable or important than otherworldly, spiritual [aspects]. It has affected people's perceptions of the environment, what is considered most important to focus on during the remaining time on this planet, so to speak. So, I would say there are reasons for why this has not been a prioritised issue, attributable to our theology, rather than science denial or denial of what is going on in the world."

This aspect was also elaborated on by the Swedish Alliance Mission representative:

> "The Church has always been fighting these ancient Greek gnostic ideas, that the material word and the body are evil. Those ideas have always permeated Christian faith. All that matters on a grand scale is salvation. So, it doesn't really matter how we live our lives, or what we do to this planet. Those ideas linger on, actually. We still need to fight them, showing people that how we live and how we care for this world matter too, because it's God's creation."

Another connected explanation brought up during the interviews related to some assembly members' views on the end times. For some assembly members, especially elderly members, given that the world will come to an end at some point in the future, there are more important issues to be addressed, such as saving people's souls (see below).

A third possible explanation was related to the assembly members' political convictions. The Swedish Alliance Mission's representative mentioned that, for some members, environment and climate issues could be perceived as 'left-wing' issues that should not be part of the Church's work.

Several informants emphasised that climate change and sustainable development are integrated parts of their international development work, particularly through Agenda 2030, which lays the foundation of international development work implemented in coordination with the Swedish International Development Cooperation Agency (SIDA). Moreover, the informants did not consider climate change and poverty alleviation as conflicting; conversely, they saw a direct link between changes in climate and poverty. One of the Pentecostal Alliance representatives argued:

> "Today, working with environmental issues is the same thing as working with poverty relief. In many countries, it is the prime driver of poverty or increase in poverty. Crop failures are caused by drought. If you can develop strategies to mitigate a dryer climate, then you will automatically combat poverty. That's the attitude you must have."

### 4.6. Cost–Benefit Considerations

Cost–benefit considerations were not given so much attention in the interviews. The idea that the cost of mitigating measures would exceed the resulting benefits was generally not shared by the denominations: it was not seen as a morally relevant aspect to focus on, which might explain the shortage of data for this argument.

### 4.7. Apocalyptic Views and Responsibility

Several informants stressed that their denominations were historically eschatological movements for which Jesus Christ's second coming and the resurrection of the world were central theological themes. When asked how perceptions about the end times could

potentially affect the assembly members' willingness to participate in pro-climate activities, the informants again emphasised the heterogeneity of beliefs within their denominations. To some degree, beliefs about the end times were stated to be correlated with age, with older members generally being more prone to believing in God's total destruction of the present world and creation of a new order, whereas younger members were generally more prone to believing in God's redemption and a continuation of the present world, although in a different state. Several informants agreed that, to some extent, one's interpretation affects the willingness to participate in pro-climate activities. One of the Evangelical Free Church's representatives argued:

> "We are supposed to propagate the kingdom of God until he returns. If you believe that the earth and heavens will burn and that God will create a new universe somewhere, or if you believe that [ . . . ] the creation will be perfected once Jesus Christ returns . . . many believe in the latter, then it becomes much more natural to care for the environment."

The Swedish Alliance Mission representative stated:

> "There are certain groups emphasising Jesus Christ's second coming and the destruction of this world through fire. For sure, this will affect their incentives to care for the creation. And then there is a group of people, to which I happen to belong, who believe that Jesus Christ will return to redeem this world. That's how I interpret the Bible. I believe that, regardless of how you interpret the Bible, you have a responsibility to care for the world as it is: what I do today will have some eternal value."

One of the Pentecostal Alliance's representatives confirmed that there is an ongoing discussion about, and also a change in attitudes regarding, the interpretation of the end times narrative:

> "You look upon [the creation] as something that we are destroying. And the change in our eschatology . . . among members of the Swedish Pentecostal movement, when you read about what is going to happen in the future you realize that a new heaven and earth . . . that from an exegetical point of view it's about a redeemed heaven. You realize that there is much greater continuity between the world that we presently inhabit and what will be in the future than previously thought. Before, people told themselves: 'Well, this is going to burn anyway . . . life will continue elsewhere'".

Regardless of their interpretation, it was clear from the informants' statements that the Apocalypse could in no way be used as an excuse not to care for God's creation. The Seventh-day Adventist Church representative stated:

> "We believe in the Apocalypse. We believe that the world as we know it is coming to an end. We don't believe that humans can avert the disaster we are facing. We don't know when this will happen, but as Adventists our focus is really not on the Apocalypse but rather on Jesus Christ's second coming. That he will return and redeem his creation. We are definitely an apocalyptic movement in that sense, but we're not a pessimistic apocalyptic movement. And we're not an apathetic apocalyptic movement."

## 5. Discussion and Conclusions

Our results show that the Swedish evangelical organisations' positions on climate change are well in line with the creation care movement and the eco-theological positions expressed through the work of the EEN and the ECI and do not at all resemble the Cornwallists' beliefs and values. This can be seen both in the pronounced ambition to work towards emissions reduction expressed through various policy documents and in the interview data, as well as in the theological justifications for carrying out this ambition. Below, we critically discuss these findings individually, drawing on previous literature. We then discuss the

relevance of the combined findings in a broader socio-historical perspective, sketching what we believe are the most important implications for future research.

*Climate science denial*: From the interviews, it is clear that the Swedish denominations oppose the Cornwallists' views on climate science, as well as those of the previous Trump administration. With the possible exception of the Seventh-day Adventist Church, which argued that climate change can partly be attributed to humans having allowed evil to reside in the world, trend, attribution and impact denial, as defined by Rahmstorf (2004), do not exist within the denominations, aside from some (usually elderly) members in the local parishes. On the contrary, all denominations take climate change science seriously. They are generally deeply concerned about the anticipated effects of climate change and consider their organisations to have a strong moral responsibility to take mitigation and adaptation measures, in particular to support people and communities living in poverty to cope with climate change. Hence, the prevailing standpoint of the interviewed denominations on climate change are much in line with the views of the NAE (2018), namely that climate change is 'a threat multiplier' that has a number of undesirable consequences, ranging from a rise in sea levels and flooding to hot temperature extremes, drought and other extreme weather events, negatively impacting people's livelihoods and well-being around the world.

*Stewardship* vs. *paganism*: Our results do not confirm Fowler's finding (Fowler 1995, p. 46) that some fundamentalist Protestants perceive the environmental movement as "the prevailing cult of our time" (see also Simmons 2009). Nothing in the studied policy documents or in the interview data suggests anything remotely resembling the Cornwall Alliance's 'Green Dragon' rhetoric (Hempel et al. 2014). The denominations were not particularly concerned about the risk of paganism or that environmentalism among their members would compete with and ultimately replace their faith in Jesus Christ. Additionally, they did not see climate measures or policies as necessarily breaking the hierarchy between God and man by elevating either "the environment to an equal plane with God" or "the environment above or on par with humans" (Peifer et al. 2014, p. 383).

However, some interviewees recognized that environmental engagement could go too far, just as any other personal efforts or ambitions, suggesting that there is still room to investigate the perceived boundaries of climate action in an evangelical Christian perspective. It could be the case that if there is any noticeable division in climate opinion between evangelicals and secular individuals and groups in a country such as Sweden, this division will be over possible solutions, rather than the basic facts of climate science or the perceived need to care for God's creation. This would be in line with the evangelical atmospheric scientist Katharine Hayhoe's conclusion: "I have never met a single human, in thousands of conversations I've had, who truly had a problem with the idea that we are to be good stewards of God's creation, and to care for those less fortunate than us [ . . . ] Almost every person I've met who had objections to climate change, had an objection to what they perceived to be the solutions." (cited in Stover 2019, p. 69).

*The dominion injunction*: The denominations place themselves quite close to each other and near the EEN and the ECI's evangelical definition of stewardship (Peifer et al. 2014). They apply a 'duty model of stewardship', in which humanity is a part of but also superior within the creation, and thus has a special role of responsibility and "service, servanthood, and caring as modelled by Christ" (Fowler 1995, p. 33). The eco-theological basis for this stewardship duty is two-fold, as suggested by Perez Sheldon and Oreskes (2017). Creation care through emissions reduction is morally called for both as a means of protecting God's creation, which is good in itself, and as a means of protecting (poor) people. In this respect, our informants position themselves close to the views developed by the early evangelical eco-theologians.

The disagreement over the meaning of the dominion injunction that exists in the U.S. was not visible among the interviewed Swedish denominations. Wardekker et al. (2009) distinguished between 'conservational stewardship', 'developmental stewardship' and 'development preservation'. Conservational stewardship means caring for the creation as God made it, while developmental stewardship includes an injunction to enrich creation, for instance, through the development of natural resources, 'turning wilderness into a garden' in Cornwallist terminology. Development preservation aims to strike a balance between the two. From our interview data, it was not possible to exactly position the denominations on the scale between conservational and developmental stewardship, although it was clear that all informants who elaborated on the matter considered the dominionism advocated by Beisner (1997) and the Cornwall Alliance morally problematic.

*God's omnipotence*: The interviewed denominations clearly agreed with the position that God is omnipotent. However, instead of taking God's perceived omnipotence as an excuse for doing nothing, all informants emphasised the humanity's responsibility to mitigate climate change. None of the denominations understood such actions to be contrary to God's plan or an insult to his divine powers. Thus, the position that it is outrageously arrogant of humans to believe that they, in the words of U.S. Senator James Inhofe, "would be able to change what He is doing in the climate", is not embraced by the Swedish evangelical denominations (Veldman 2019, p. 2).

One informant framed the responsibility of humans to care for God's creation in terms of 'co-creatorship', meaning that humans have an active part to play in God's continuous creation. On the face of it, this appears to be consistent with how the Cornwallists understand the role of humans within creation, namely as 'enrichers' of God's creation. However, the co-creatorship of humans elaborated on by our informant had little to do with enriching the creation along Cornwallist lines and more to do with honouring God's intentions and efforts by respecting certain limits and finding a balance in preserving and making use of God's creation.

*Moral relevance of climate change*: One of the most forcefully voiced arguments against emissions reduction put forward by the Cornwallists is that such measures would seriously hamper efforts to combat poverty (Hempel et al. 2014). Thus, strong moral reasons speak against implementing emissions reduction. However, according to our informants the opposite holds true: mitigation and adaptation are morally required, not least from a social justice perspective. They did not see any conflict between taking climate measures and combating poverty. The view that climate change mitigation would not be a genuine Christian responsibility, in contrast to poverty alleviation, was considered illogical and morally indefensible by our informants.

*Apocalyptic views*: Through fieldwork conducted among evangelicals in Colorado Spring, Bjork-James (2020) explored two contrasting interpretations of the end times. According to the most frequently held view, the end times involve a complete destruction of the Earth and a relocation of the faithful "to a distant and pristine heaven" (p. 3). According to the alternative narrative, God will not destroy the Earth but instead redeem it, creating a new heaven on Earth. The second view was far less common among the evangelicals in Bjork-James' study; however, this view is gaining ground among younger generations of evangelicals. Our research data suggest that a similar division of views exists within the Swedish evangelical denominations, with some members adhering to the former, traditional interpretation and others supporting the latter. As noted above, at least one of our interviewees explicitly endorsed the second interpretation. However, regardless of which interpretation is most common within the respective denomination, it was clear from our interview data that none of our interviewees considered 'end times climate apathy' to be justified: just as God's omnipotence cannot justify inaction, the fact that the world will one day come to an end is not a reason to refrain from doing what we can to care for God's creation.

In summary, our study indicates that there is little evidence to suggest that the climate-oppositional views and arguments of the Cornwallists have been taken up by the Swedish

evangelical community. The lack of support for Cornwallist views and the observed unity of positions on climate change among the Swedish evangelical denominations bring to the fore several questions.

Firstly, what factors can help to explain the observed lack of Cornwallist views among Swedish evangelical denominations? As noted above, biblical literalism and political conservatism were found to be associated with a lower degree of concern for the environment. Do Swedish evangelicals hold less literalist views of the Bible than their U.S. counterparts? Alternatively, are they less politically conservative?

From our data, it is clear that the participating denominations certainly do not ascribe to Cornwallist dominionism, nor do they worry that support for environmental or climate policy would lead to paganism among their members. However, in many respects they appear to adhere to a literalist interpretation of the Bible. The Bible is considered important in dictating how to live their daily lives. This is perhaps most pronounced in their views of salvation and God's omnipotence, and in their eschatology. Thus, it is not obvious that the support for environmental and climate policy embraced by the participating denominations can be explained by reference to a lesser degree of biblical literalism.

Political orientation could, to some extent, explain these observed differences. Voting studies show that both American and Swedish evangelicals tend to vote for right-wing parties. In Sweden, support for conservative politics, represented by the Swedish Christian Democrats, has been strong since the party was founded in the 1960s (Hagevi 2008). However, before this, members of the so-called 'free churches' in Sweden tended to vote for either the Swedish Social Democratic Party or, in particular, the Liberal Party, both of which are ideologically closer to the U.S. Democratic Party than the Republican Party (Hagevi 2008). In the late nineteenth century, Swedish evangelicals formed an alliance with social democrats and liberals to push for a modern welfare state with democratic rights and a redistributive economic agenda, as an alternative to the existing conservative establishment (Halldorf 2020). Thus, early on in the history of the Swedish evangelical movement, individual rights and freedoms were advocated for through the establishment of, and not in opposition to, a strong welfare state. The idea of 'freedom through the welfare state' was later on taken up by the Swedish Christian Democrats. As a consequence, although considered conservative, the positions of the Swedish Christian Democrats differ from those of the U.S. Republican Party in important respects (Halldorf 2020). Political orientation was not specifically discussed in our interviews. We believe that further studies are needed to establish how political and theological beliefs and values interact in the formation of positions on climate change within the evangelical community.

A related route for exploring the noted differences between the Swedish and U.S. evangelical communities on the issue of climate change could be through the history of Swedish evangelicalism and the socio-political landscape that Swedish evangelicals have traditionally navigated (cf. Veldman 2019). Historically, Swedish evangelical communities have operated, not only in a country with a strong conservative establishment mainly granting rights to the economically privileged, but also with a strong Protestant state church. For a very long time, Swedish evangelicals were oppressed and marginalised. This is markedly different from the conditions under which evangelicalism developed in the U.S., a country to which many religiously oppressed Europeans emigrated (Wilkinson 2012; Veldman 2019). Hagevi (2008) argued that the socio-political situation in Sweden contributed to giving evangelicals in this country similar experiences to those of Black and other minority evangelicals in the U.S., something that partly helps to explain the significant degree of support for environmental and climate policy among Swedish evangelicals. We believe that this hypothesis deserves to be investigated in further detail in relation to other potential explanatory factors, including a straightforward 'creation care theological basis'.

Although evangelicals and other religious minorities in Sweden are no longer oppressed by the state, they have traditionally played a minor role in Swedish politics. Sweden has been a secular country for a long time, with religion exerting a limited influence on politics. There are few well-funded religious think tanks in Sweden, and there have, as far

as we know, been no close ties between the evangelical community, industry and politically conservative anti-government advocates in Sweden. Thus, an organised 'climate science denial machine', in the sense described by Dunlap and McCright (2011), which could pick up on potentially anti-environmental or anti-climate sentiments within the evangelical community and help incorporate them into evangelical identity, simply does not exist in Sweden.

The lack of studies on religiously motivated climate change denial outside of a U.S. context makes it difficult to firmly establish whether Sweden represents an outlier or whether the U.S. constitutes an exception to an otherwise fairly concordant international evangelical community. Empirical investigations of evangelical communities in other parts of the world could help to increase our understanding of the factors behind religiously motivated climate change denial, as well as the reasons behind its proliferation in the U.S. However, given that the World Evangelical Alliance (WEA), an organization that gathers evangelicals and evangelical denominations from all over the world, has adopted a progressive approach towards climate mitigation and adaptation, and recently started a WEA Sustainability Center (WEA 2022), it is perhaps not unreasonable to assume that it is the U.S. that constitutes the exception to an otherwise fairly unison international evangelical community.

*Limitations and opportunities for future research*: When discussing the present study's results, a number of limitations must be kept in mind. Firstly, it could be argued that our conclusions are not valid since we did not measure the right entity. If we want to say something about the beliefs and attitudes to climate change that exist within the evangelical community in Sweden, then investigating the formal standpoints of evangelical denominations may be misleading. As noted above, the Swedish evangelical movement (similar to evangelical movements in other parts of the world) is an anti-authoritarian and heterogeneous movement, with local parishes enjoying a great degree of freedom in developing their theological orientations. Therefore, the interview data cannot be taken to represent each and every individual member, or even each and every member church or local parish. To gain a fuller picture of the Swedish evangelical community's views, perceptions of climate change, and the need for mitigation measures, the present study would have to be supplemented with, for instance, studies of preaching manuscripts to determine what is actually being preached in local parishes or surveys investigating the views of individual church members. We hope that the present study can serve as a starting point for further empirical studies into the climate change views and perceptions of evangelical communities in Sweden and other countries.

Secondly, there is a theoretical possibility that the interviewed representatives were not entirely frank when reporting on their denominations' views on climate change. In Sweden, the dominant political and societal views on climate change lie very close to those of the IPCC. Climate science denialist standpoints certainly exist; however, they receive relatively sparse attention in established media outlets. The Swedish government's view can be characterised as 'pro-climate' and 'pro-IPCC'. This also applies to public agencies, such as the Swedish International Development Cooperation Agency, with whom several of the participating denominations have active collaborations. Questioning climate science openly could, thus, make it more difficult to maintain those collaborations, which could present a reason for the denominations to not openly share potential doubts about climate science with us. In our view, however, this is unlikely. The standpoints reported during the interviews correspond with officially published statements on the denominations' websites and are also in line with official declarations supporting climate science and mitigation measures signed by some of the denominations, such as the Cape Town Commitment (The Lausanne Movement 2011), as well as the general views on climate change science and policy embraced by the World Evangelical Alliance.

Thirdly, it might be argued that our study is based on limited empirical data, and thus our research results are not sufficiently reliable; only four evangelical denominations participated in our study. Although we were able to secure participation from a majority of the major evangelical denominations in Sweden, it would have been better if all the (seven) identified denominations had agreed to take part in our study. However, in the case of Equmenia, which is one of the largest evangelical denominations in Sweden, a quick review of the publicly available documents on climate change issued by the church shows that their position is very similar to the ones captured in our study. For the other two denominations, the Salvation Army and Word of Faith Movement/Word of Life, there is much less material publicly available. Therefore, it is difficult to know for sure what their positions on climate change are, or to what extent their inclusion in the study would have affected our conclusions.

Finally, in this paper we used the climate counter-arguments put forward by the Cornwall Alliance and associated actors in the U.S. as a departure point for our analysis. We believe that this is a reasonable departure point given the close historical ties between Swedish and American evangelicalism, although there are significant differences between the two at present, as described by Halldorf (2020) and the strong influence of Anglo-American culture in Sweden generally. However, it is important to recognize that the climate counter-narrative represented by the Cornwall Alliance and its associated actors constitutes one particular lens through which evangelical climate denial could be analysed. In other parts of the world, other counter-discourses (as well as pro-climate eco-theological positions) may exist that could help shed light on this issue.[6] Kempf (2020), for instance, underscores the significance of the Noah myth (i.e., God's promise to Noah never to send another flood) to the climate scepticism of many Pacific Islanders (see also Rubow and Bird (2016) and Fair (2018) for more detailed discussions of 'the Noah controversy'). The potential impact of the Noah myth on our denominations' climate-related positions was not explored in this paper; however, we believe that this and other dominant Christian counter-discourses developed in non-Western cultural settings can help contextualise this issue and further deepen our understanding of the complex issue of religiously informed climate change denialism.

**Author Contributions:** Conceptualization, K.E.B. and M.K.; methodology, K.E.B. and M.K.; validation, K.E.B. and M.K.; formal analysis, K.E.B. and M.K.; investigation, K.E.B. and M.K.; resources, K.E.B. and M.K.; data curation, K.E.B. and M.K.; writing—original draft preparation, K.E.B. and M.K.; writing—review and editing, K.E.B. and M.K.; project administration, K.E.B.; funding acquisition, K.E.B. and M.K. All authors have read and agreed to the published version of the manuscript.

**Funding:** This research was funded by Formas—The Swedish Research Council for Environment, Agricultural Sciences and Spatial Planning grant number 211-2014-595 ("Mind the Gap").

**Institutional Review Board Statement:** An application for ethical vetting was submitted to and approved by the Swedish Ethical Review Authority, decision 2018/796-31/5, 17 May 2018.

**Informed Consent Statement:** Informed consent was obtained from all subjects involved in the study.

**Data Availability Statement:** The transcribed interviews (in Swedish) are stored at the Division of Philosophy, KTH Royal Institute of Technology and are available on request.

**Acknowledgments:** The authors would like to thank two anonymous reviewers for their valuable comments and suggestions. The authors would also like to thank the interviewed representatives of the Evangelical Free Church, the Pentecostal Alliance, the Swedish Alliance Mission, and the Seventh-day Adventist Church for their participation in the study. Finally, the authors would like to thank Malcolm Fairbrother, Sverker Jagers, Joakim Kulin and the other participants in WS2 at the 14th Nordic Environmental Social Science Conference in Luleå, 10–12 June 2019 for their valuable comments on a previous version of the article.

**Conflicts of Interest:** The authors declare no conflict of interest.

**Appendix A. Interview Guide (Translated from Swedish)**

1. What is the denomination's views on the conclusions drawn by in the IPCC reports? That is, climate change is occurring, is anthropogenic in nature, and will have predominantly negative impacts on humans and the environment. Has the denomination adopted any policy documents on climate change?

2. A common belief among Christians is that humans have a responsibility to care for God's creation. What is the denomination's position on this? What does stewardship entail, and what are some consequences for climate policy?

3. Some Christian groups argue that it is problematic to put sustained focus on environmental and climate issues, since it may lead to people worshipping the Creation rather than the Creator. What's the denomination's position on this? Can environmental and climate engagement go too far, bordering on paganism or idolatry?

4. Some Christian groups argue that humans have a right and duty to rule over the rest of Creation and that this involves "enriching" the Creation, for example, through economic development, by extracting minerals and fossil fuels, etc. What's the denomination's position on this? Do humans have a right to use the environment in order to create welfare for its own kind? How should conflicts between human welfare and the welfare of other species/the environment be balanced?

5. Some Christian groups argue that, since God is omnipotent, he will solve the climate problem (if it even exists). What's the denomination's position on this?

6. Some Christian groups argue that environmental and climate issues are inconsequential from a moral point of view and that Christians should focus on other issues such as abortion, gay marriage and poverty alleviation instead. How important is the climate issue relative to other social and political issues?

7. Some Christian groups argue that it is more cost-efficient to spend money on poverty alleviation than to reduce our greenhouse gas emissions, the latter of which may benefit only future generations of humans. What's the denomination's position on this?

8. Some Christian groups argue that the Apocalypse, as depicted in the Book of Revelation, will take place sometime within the coming years and that this gives us a reason not to take the climate issue seriously. What's the denomination's position on this?

9. Is there anything else that the denomination would like to add?

**Notes**

[1] See Wilkinson (2012) and Veldman (2019) for detailed expositions of how the U.S. evangelical community has historically addressed climate issues.

[2] The Cornwall Alliance developed this point further in the 'Resisting the Green Dragon' series (Hempel et al. 2014).

[3] Since then, the Evangelical Free Church and the Pentecostal Alliance of Independent Churches have terminated their memberships of the Swedish Evangelical Alliance.

[4] Decision 2018/796-31/5, 17 May 2018.

[5] The transcribed interviews (in Swedish) are available on request.

[6] We would like to thank an anonymous referee, who pointed this out.

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
