# Peer review of "Faithful Stewards of God’s Creation? Swedish Evangelical Denominations and Climate Change"

_religions, doi:10.3390/rel13050465_

Round 1

Reviewer 1 Report

Review of

Faithful Stewards of God’s Creation? Swedish Evangelical Denominations and Climate Change

For

Religions

General Comments

This is a fine research article that presents new findings that extend our understanding of the nexus between faith and climate change action in Sweden’s evangelical community. It deserves to be published. The report is nuanced, well supported by literature, and reports a reasonable methodological approach to address the study’s stated aims. I am suggesting some revisions that the authors should consider implementing in order to expand and enhance what is already a very good research article. More specifically, I am suggesting you consider some of the following was to additionally strengthen your article:

The publication may gain a bit more strength by engaging with some literature from other parts of the world (e.g., the South Pacific), which presents a markedly different perspective on the ideology, history, and experience of the United States. Some literature is suggested below for possible consultation by the authors. Engaging with some of this literature would show that the US Cornwallists’ perspectives are not the only counter-reference point of global influence, but that there are other positions (and other interpretations of scripture and environmental change) that point to a more pro-climate action agenda, both by churches and state actors. The authors need not necessarily engage with all of these positions as their framework comparison against the Corwallists is indeed lucid and helpful to structure their argument. However, I believe it would be a missed opportunity not to at least acknowledge some other paradigmatic perspectives on ecotheology, including from the Pacific. To that end, minor revisions will make the article even stronger and more balanced for an international readership. This is important to keep in mind as readers of the journal will be from all around the world. Incidentally, this is also worth noting in relation to the article’s discussion of some references to Sweden’s history (Lines 706-707: “Historically, Swedish evangelical 706 communities have operated in a country with a strong Protestant state church. For a very 707 long time, they have been oppressed and marginalised.”) Most international readers will be entirely unaware of this history and how it has shaped the current expression of the Swedish church. For this reason, the authors should make some references to the unique history of the Swedish church at large (see Specific Comments below) with one or two explanatory notes or footnotes. Finally, there is also a sense that the article could benefit from a bit more literature on both Eschatology and Paradigmatic Perspectives on Christian climate action and hermeneutic. Once again, some literature is suggested below for possible consultation by the authors. Notwithstanding these suggestions, it bears repeating that the article is clear, compelling and stands out as one of the very few empirical studies conducted in this area of research, which - despite relying on a small sample - has convincingly presented a very helpful synthesis of Swedish evangelical perspectives on an important contemporary issue. These minor revisions are recommended to further improve the article.

Specific Comments:

Sections 1. Introduction and 2. Climate science and the evangelical community

line 52: “Most research on evangelicalism and climate science and policy has focused on U.S. denominations.” - This is true, however, it would be a missed opportunity not to at least mention some of the innovative conceptual and empirical research that has been done elsewhere, e.g., in the Pacific/Oceana. (See above, General Comments; see some lit below)

line 60: surely must mean “Sweden” instead of “Swedish” - correct.

lines 61-65: “These and similar findings point to the need 61 for investigating the relationship between evangelicalism and climate science and policy 62 in different countries, also outside of the U.S. Such studies could help to contextualise the 63 phenomenon of climate denial and, by extension, shed light on the factors behind faith- 64 based climate opposition in countries such as the U.S.” (as mentioned, some additional examples/references would better contextualise the discussion).

lines 76-78: “By addressing those questions, we hope to contribute to the sparse 76 corpus of qualitative studies investigating the relationship between evangelical faith and 77 attitudes to climate science and policy outside of a U.S. context.” (same here, some more lit.)

lines 107-110: “Veldman (2019) argues that evangelical climate denial in the U.S. cannot be under- 107 stood in isolation from the socio-historical context in which it developed. For an extensive 108 period of time after the nation was founded, evangelicals had considerable political influ- 109 ence through their key positions in society.” (Observation: Very valuable discussion in this section! - no change suggested, just noting the very helpful framing of the article!)

lines 137-139: “However, further studies into the 137 theological drivers of climate opposition are required to better understand how the reli- 138 giously-based climate oppositional groups position themselves and are able to sell their 139” (see comment about some more lit. mentioned above)

Line 252: “A forth argument” - correct: “fourth”.

Section Methods

Line 277: “2. Materials and methods” - please correct: should be “3. Materials and methods”.

Lines 327-328: “The template was communicated to the denominations 327 beforehand to give them the opportunity to prepare for the interviews. The interviews 328 lasted for 45–60 minutes and were recorded and transcribed.” - I think it will help the reader to learn a bit more about how the template was developed (more specifically), including some of its formats. The authors may also consider including the questionnaire as an Appendix file in this research or at the very least mention some example questions and areas of interview interest. In this way, the reader has a better awareness of the interview process. In expanding this part, I would also suggest that the authors strengthen this section by arguing (with specificity and justifications) why the comparatively small sample (seven interviewees) is still to be considered to return valid and reliable data/findings. 

Section Results

Line 335: “3. Results” - Correct to: “4. Results”

There are a lot of helpful verbatim quotes that should be indicated as such more clearly by inserting quotation marks at the start and end. On occasion, there are quotes within quotes. As a reader, my preference would be to reflect longer verbatim quotes by formatting them as so-called “block quotes” (indenting them). If the journal style does not allow this then at the very least the article needs quotation marks rather than merely a new paragraph. It just makes the reading so much simpler. Where there are quotations within quotations, single quotation marks should be placed within double quotation marks (“… ‘…’ …”). Here are the line numbers where this should be done. Readers will be immensely grateful for making this copyeditorial revision:

Lines 368-370; 376-377; 383-389; 397-402; 410-414; 427-430; 433-438; 458-461; 468-473; 475-480; 493-499; 501-506; 522-525; 545-549; 550-555; 559-565; 569-574; 

Short subsection 4.6: 

See my reference above for information about the interview guide/template. The reader is left unsure whether the absence of data arose from not asking for such information. In any case, even if there were no such questions asked, this could be mentioned in a section called “Limitations and Opportunities for Future Research”. Such a section would wonderfully create space for some of the excellent questions raised by this research, e.g., lines 684-685: “Do Swedish evangelicals hold less literalist views of the Bible than their U.S. 684 counterparts? Alternatively, are they less politically conservative?”

Lines 706-708: Please see my earlier point about explaining Sweden’s history for an international readership. Overseas readers will be immensely grateful for a couple of explanatory sentences added here: “Historically, Swedish evangelical 706 communities have operated in a country with a strong Protestant state church. For a very 707 long time, they have been oppressed and marginalised.” And here: “Although evangelicals and other religious minorities in Sweden are no longer op- 717 pressed by the state, they have traditionally played a minor role in Swedish politics.”

Lines 727-732: Once again, I would suggest adding a section entitled Limitations and Opportunities for Future Research to more clearly take stock of the article’s research contribution to knowledge, and to sketch opportunities for how others may further develop the field going forward. A short paragraph/subsection would be immensely helpful and could comprise and expand on some of the information already discussed, e.g., here: lines 727-732; 750-751; 783-787.

Once again, it was a delight to review this article. Thank you for the privilege. Friendly colleagial wishes for your current and future work.

References

Fair H (2018) Three stories of Noah: navigating religious climate change narratives in the Pacific Island region. Geo Geogr Environ 5(2). https://doi.org/10.1002/geo2.68

Griffiths, J.D. (2021) Wonders in the Heavens Above, Signs on the Earth Below: Pacific Islands Pentecostalism, Climate Change and Acts 2. In Luetz, JL & Nunn, PD (Eds.) Beyond Belief—Opportunities for Faith-Engaged Approaches to Climate-Change Adaptation in the Pacific Islands (pp. 329-344), Springer, https://doi.org/10.1007/978-3-030-67602-5_17

Havea PH, Hemstock SL, Des Combes HJ, Luetz JM (2018) “God and Tonga are my inheritance!”— climate change impact on perceived spirituality, adaptation and lessons learnt from Kanokupolu, ‘Ahau, Tukutonga, Popua and Manuka in Tongatapu, Tonga. In W. Leal Filho (eds) Climate change impacts and adaptation strategies for coastal communities. Springer Nature, Switzerland, pp 167–186. https://doi.org/10.1007/978-3-319-70703-7_9

Orr DW (2005) Armageddon versus extinction. Conserv Biol 19(2):290–292. https://doi.org/10.1111/j.1523-1739.2005.s04_1.x

Luetz, J.M., Buxton, G., & Bangert, K. (2018). Christian Theological, Hermeneutical and Eschatological Perspectives on Environmental Sustainability and Creation Care—The Role of Holistic Education. In J.M. Luetz, T. Dowden, & B. Norsworthy (Eds.), Reimagining Christian Education—Cultivating Transformative Approaches (pp. 51–73). Springer Nature, Singapore. https://doi.org/10.1007/978-981-13-0851-2_4 

Ernst M (2012) Changing Christianity in Oceania: a regional overview. Archives de sciences sociales des religions. http://doi.org/10.4000/assr.23613

Ernst M, Anisi A (2016) The historical development of Christianity in Oceania. In Sanneh, L. & McClymond, M.J. (Eds.), The Wiley Blackwell Companion to World Christianity, 588–604, Wiley, https://doi.org/10.1002/9781118556115.ch44

Ayre C (2014) Earth, faith and mission: the theology and practice of earthcare. Morning Star Publishing, Northcote, Vic

Nunn PD, Mulgrew K, Scott-Parker B, Hine DW, Marks ADG, Mahar D, Maebuta J (2016) Spir- ituality and attitudes towards nature in the Pacific Islands: insights for enabling climate-change adaptation. Clim Change 136(3–4):477–493. https://doi.org/10.1007/s10584-016-1646-9

Kempf W (2017) Climate change, christian religion and songs: revisiting the Noah story in the Central Pacific. In: Dürr E, Pascht A (eds) nvironmental Transformations and Cultural Responses. Palgrave Macmillan, New York, pp 19–48

Kempf W (2020) Introduction: climate change and Pacific Christianities. Anthropological Forum 30(3):215–232. https://doi.org/10.1080/00664677.2020.1812052

Rubow C, Bird C (2016) Eco-theological responses to climate change in Oceania. Worldviews 20:150–168
